# Ionic imbalance induced self-propulsion of liquid metals

Ali Zavabeti[1], Torben Daeneke[1], Adam F. Chrimes[1,2], Anthony P. O'Mullane[3], Jian Zhen Ou[1], Arnan Mitchell[1], Khashayar Khoshmanesh[1] & Kourosh Kalantar-zadeh[1]

Components with self-propelling abilities are important building blocks of small autonomous systems and the characteristics of liquid metals are capable of fulfilling self-propulsion criteria. To date, there has been no exploration regarding the effect of electrolyte ionic content surrounding a liquid metal for symmetry breaking that generates motion. Here we show the controlled actuation of liquid metal droplets using only the ionic properties of the aqueous electrolyte. We demonstrate that pH or ionic concentration gradients across a liquid metal droplet induce both deformation and surface Marangoni flow. We show that the Lippmann dominated deformation results in maximum velocity for the self-propulsion of liquid metal droplets and illustrate several key applications, which take advantage of such electrolyte-induced motion. With this finding, it is possible to conceive the propulsion of small entities that are constructed and controlled entirely with fluids, progressing towards more advanced soft systems.

[1] School of Engineering, RMIT University, Swanston Street, Melbourne, Victoria 3001, Australia. [2] Institute of Chemical and Bioengineering, Department of Chemistry and Applied Biosciences, 8093 ETH Zürich, Switzerland. [3] School of Chemistry, Physics and Mechanical Engineering, Queensland University of Technology (QUT), Brisbane, Queensland 4001, Australia. Correspondence and requests for materials should be addressed to K.K.z. (email: kourosh.kalantar@rmit.edu.au) or to T.D. (email: torben.daeneke@rmit.edu.au).

**S**oft components based on microfluidic and elastomer technologies present an increasing promise for industrial uptake[1–12] and relatively non-hazardous liquid metal alloys of gallium are set to play a considerable role in this process[13]. Room-temperature liquid metals have shown to be remarkable platforms for makeshift mechanical components[14–16], reversible electrochemical systems[17,18], soft sensors[7,17,19,20], electrical components in microfluidic channels[21], three-dimensional printing[22,23] as well as stretchable and reconfigurable electronics[24–26]. Controlling the motion and deformation of liquid metals is the key to the successful realization of these applications. To date, motions and deformations of liquid droplets have been demonstrated using methods such as surface oxidation modifications[27,28], surfaces that are selective to vapours[29], hydrodynamic modifications[30], topologically modified structures[31,32], fuel consumption[33,34], provision of energy through light sources[35], electrical energy sources[18,27,36–42] and magnetic fields[43].

In principle, the applied driving force should break the charge symmetry that exists on the surface of a liquid metal to generate a differential pressure that either changes the configuration of the liquid and/or induces a displacement. However, there are no studies that show this symmetry can be broken solely due to differences in the environment surrounding a liquid metal. Specifically, studies to date have not focused on the significance of the ionic composition near the surface of the metal, where an electrical double layer (EDL) forms on the droplet surface giving rise to surface capacitive properties[44], which can be used for modifying its behaviour.

The ionic imbalance at the interface between the liquid metal and the solution approaches equilibrium through the formation of an EDL[45]. For a liquid metal immersed in an ionic solution, the EDL can be modelled as a parallel-plate capacitor[18]. Based on the integrated Lippmann's equation, the surface tension of the liquid metal droplet $\gamma$ changes with the square of the potential as[46]:

$$\gamma = -\frac{C}{2}(\varphi - \varphi_0)^2 + \gamma_0 \qquad (1)$$

in which $C$ is the EDL capacitance per unit area, $\varphi$ is the electrode potential, $\varphi_0$ is the potential of zero charge (PZC) and $\gamma_0$ is the maximum surface tension at PZC. Knowing $\gamma$, the Young–Laplace equation can be used for defining the pressure difference ($\Delta P$) across a liquid metal hemisphere as[44]:

$$\Delta P = \gamma \left(\frac{1}{R_1} + \frac{1}{R_2}\right) \qquad (2)$$

where $R_1$ and $R_2$ are the principal radii of curvature at the interface, respectively. Equations (1) and (2) describe the change that the surface tension provides as a means to alter the pressure difference that in turn can produce a displacement and/or cause deformation of the soft liquid metal.

Changes in surface tension may be induced by directly adjusting the EDL, via modifying the electrolyte surrounding the droplet that generates a surface tension difference. One possible change to the electrolyte that would affect the EDL is adjusting the ionic content by altering the pH of the solution. Another possibility is to change the composition of the electrolyte itself. A change of electrolyte has been previously shown to affect the surface tension for mercury[47] and liquid gallium[48]. It was demonstrated that the PZC, which corresponds to the maximum on the surface tension curve, is a strong function of electrolyte type. The primary reasons for this electrolyte dependency include the variation in the electronegativity of the ions, their mobility and intrinsic charge. In addition, this dependence of the surface tension on the electrolyte composition is augmented, reaching a

maximum value, if a positive voltage is applied to the liquid metal against a reference electrode[47].

In this work, we present the mechanical actuation of a liquid metal through modification of the liquid electrolyte surrounding it. Galinstan, which is a eutectic alloy of gallium, is used as the model liquid metal. Gallium has low toxicity, negligible vapour pressure and is relatively safe for practical applications[13]. Gallium itself melts above room temperature at 29.8 °C; however, when combined with other metals its melting point can be significantly lowered to below 0 °C such as is the case with Galinstan, which is a eutectic alloy of 68.5% gallium, 21.5% indium and 10% tin[44]. The pH of the electrolyte is modified by adding acidic or basic solutions and the ionic properties of the electrolyte are adjusted through the addition of a salt. We show that maintaining a gradient in the electrolyte properties across a liquid metal droplet results in continuous mechanical motion and deformation. The device is characterized to determine the required conditions for self-propulsion and the ionic imbalance requirements are optimized and tested. Two key applications that illustrate pumping and switching effects are also presented. This work represents the first steps in building a self-propelling liquid metal droplet, which can be controlled with a fluid only, thereby marking a significant advancement towards truly autonomous soft systems.

## Results

**pH imbalance.** The first experiments were conducted to observe the dynamics of a liquid metal when two different electrolytes of varying pH were placed on each side of the droplet. An open-top fluidic channel was fabricated by milling polymethyl methacrylate, as shown in Fig. 1a. A liquid metal droplet was placed in a spherical recess located in between two different flowing electrolytes of acidic and basic nature (dyed in different colours for easy identification—very low concentration food dyes). This liquid metal droplet (3 mm in diameter) nearly filled the recess, effectively reducing the mixing between the two electrolytes. Some of the effects of droplet size variations are presented in Supplementary Fig. 1. The set-up was designed in a way that the effect of electrolytes on the droplet, causing symmetry breaking, could be readily observed using a camera. The camera recorded the presence of deformation or surface flow during the experiments, which was the main information that was required for this study.

Constraining the droplet in the recess allows one to study its deformation and surface flow dynamics without any displacement. A continuous flow of electrolyte was important, as it provided a fresh supply of ions to the liquid metal surface and cleansed the surroundings of the droplet from any undesired chemical byproducts and bubbles. Flow rates were set to a low value of 200 μl min$^{-1}$ in each channel, to ensure a laminar flow with a Reynolds number of 2.2. This value was sufficiently low to avoid mixing of the electrolytes within the two channels. More detail on materials and methods are available in the Methods section.

**Droplet dynamics under pH imbalance.** The diagram in Fig. 2 represents the droplet dynamics when a liquid metal droplet is placed in between acidic (HCl) and basic (NaOH) electrolytes. Experiments were conducted at varying concentrations of NaOH and HCl, as illustrated in Fig. 2c. Droplets were observed to have two dynamics: deformation and Marangoni flow.

To trigger a change in the dynamics of a droplet, a minimum pH difference across the liquid metal of ~13 was required. Therefore, during these experiments each hemisphere of the droplet was exposed to a minimum of 0.3 mol l$^{-1}$ NaOH

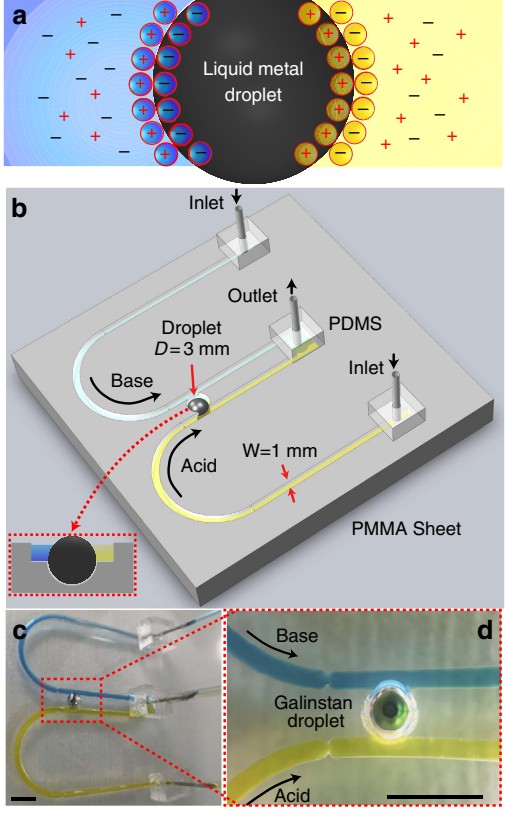

**Figure 1 | Framework for analysing liquid metal droplet dynamics under ionic imbalance.** (**a**) Top view schematic of the droplet and arrangement of ions, forming the EDL. (**b**) Schematic of the experimental setup showing two U-shaped open-top (see inset) polymethyl methacrylate (PMMA) inlet channels, which extend in parallel and join at an outlet. Two channels carry different types of electrolytes represented in distinct colours, acidic in yellow and basic in blue. Two parallel flows come in contact with the Galinstan droplet of 3 mm diameter residing in a recess. (**c**) Actual experimental set up. (**d**) Close-up view of **c**. Scale bars, 5 mm.

(pH $\sim 13.5$) and 0.3 mol l$^{-1}$ HCl (pH $\sim 0.5$), respectively. Above 3 mol l$^{-1}$, significant changes in the chemistry of the system was observed and the functionality was disrupted.

**Deformation of liquid metal droplets.** The first dynamic component was due to the deformation (extension and compression) of the liquid metal droplet at the interface with the electrolyte, as described by equation (2). Deformation was observed as a change in shape of the droplet, indicating a pressure imbalance between the two opposing hemispheres of the droplet (Fig. 2a). Each hemisphere formed a different curvature corresponding to its interfacial pressure deviation exposed to each of the electrolytes. As the flow was equal on both sides, there was no pressure deviation induced by electrolyte flow. Thus, the pH imbalance induced the pressure deviation on the two sides of the droplet, which resulted in a displacement of liquid metal from the HCl side (high surface tension) towards the NaOH (low surface tension) electrolyte. As the droplet movement was restricted by the recess, the liquid metal deformed towards the NaOH channel leading to an extended diameter ($D_1$) in one direction and a compressed diameter ($D_2$) in the other axis perpendicular to the direction of extension. We refer to the aspect ratio between the two axes as 'deformation ratio', $\frac{D_1}{D_2}$. Measured deformation ratios in Fig. 2b varied from 1 to 1.46.

**Marangoni flow of liquid metal droplets.** The second observed dynamic component was due to the Marangoni flow effect. Marangoni flow was observed as the mass transfer of liquid on the surface of the droplet, which was driven by the induced surface tension gradient[49] (Fig. 2b). The flow induced by the ionic imbalance was initiated from the interfacial region with lower surface tension (NaOH) towards the region with higher surface tension (HCl). The surface Marangoni flow path was observed along the outer curvature of the surface of the droplet outside the electrolyte (Fig. 2b). Marangoni flow rate ranges during the pH experiments in Fig. 2 were found to span from 0 to a maximum of 1.74 mm s$^{-1}$. Some small micro particles were added into the electrolytes to facilitate the observation of the Marangoni flow for velocity measurements as per our previous work[21]. The conditions that define the Marangoni flow dynamics of a droplet are presented in Fig. 2.

An important observation is regarding the formation of thin flakes and their apparent effect on the Marangoni flow. At relatively high acidic and basic concentrations, it is seen that oxide flakes of triangular configurations are formed on the surface of the liquid metal droplet (Supplementary Fig. 2a). These flakes increasingly become thicker when the concentration of the electrolyte increases (almost four to five times thicker when NaOH or HCl molar concentration increase by an order of magnitude according to Raman spectroscopy assessments, as presented in the Supplementary Fig. 2b). It seems that when these flakes (which are made of hydrated oxides of gallium) become thicker, they eventually delaminate into the electrolyte or move along together with the Marangoni flow (Supplementary Fig. 2c). The presence of the thick hydrated oxides seems to be an important reason for the dominance of Marangoni flow and reduction of the deformation effect. The flakes form a solid skin (either attached or delaminated) that contains the droplet and reduces the deformation effect.

**Dynamics regions.** The reference diagram shown in Fig. 2d summarizes the dynamics of the droplets, according to which the diagram can be divided into three regions. Region 1 features experimental conditions that resulted in droplet dynamics dominated by deformation. The droplet was found to strongly stretch towards the NaOH electrolyte with deformation ratios of up to 1.46, while Marangoni flow was less noticeable ($<0.2$ mm s$^{-1}$). The experimental conditions resulting in region 1 type dynamics were found to require comparatively low concentrations of one of the electrolytes in which either hemisphere was exposed to ionic concentrations of 0.3 to 0.7 mol l$^{-1}$ (pH $\sim 0.5$ to $\sim 0.2$ and $\sim 13.5$ to $\sim 13.9$ for HCl and NaOH, respectively) (Fig. 2c). Region 2 summarizes the experimental conditions for which a droplet exhibited simultaneous measureable deformation and Marangoni flow (Fig. 2c). As per Fig. 2d, the region is featured as curved and confined areas defined by the concentrations of NaOH between 0.7 and 1.7 mol l$^{-1}$ (pH $\sim 13.9$ to $\sim 14.2$) and HCl between 0.7 and 2 mol l$^{-1}$ (pH $\sim 0.2$ to $\sim -0.3$). Region 3 is the area dominated by Marangoni flow (Fig. 2c). Increasing both NaOH and HCl concentrations to above 1.7 and 2 mol l$^{-1}$ (pH $\sim 14.2$ and $\sim -0.3$ for HCl and NaOH, respectively) (Fig. 2d), respectively, enhances the induced Marangoni flow (up to a maximum 1.74 mm s$^{-1}$), whereas reducing the deformation ratios (to a maximum of 1.1).

Altogether, the results show that a low differential pH results in deformation towards the basic solution, whereas a large differential pH induces high Marangoni flows towards the acid solution, which counters the deformation process.

It seems that the deformation process can be repeated for a substantial time period (we repeated it for more than a week with

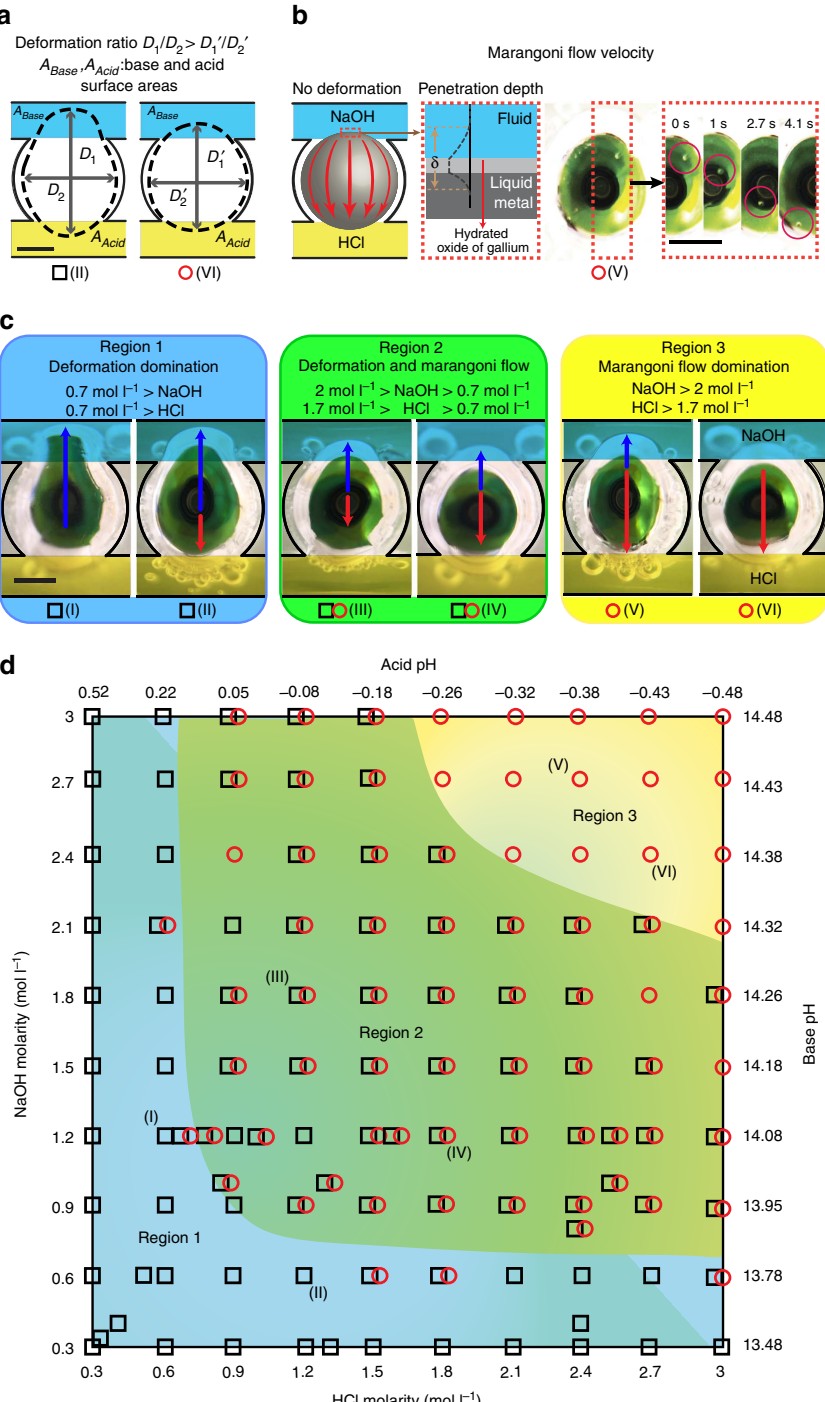

**Figure 2 | Dynamics of the liquid metal droplets under different HCl and NaOH concentrations.** (**a**) Schematic of the deformation ratio measurements for $D_1/D_2$ assessment. 'Black square' indicates experiments with deformation dominating the dynamics and Marangoni flow dominant experiments are represented by 'red circles'. (**b**) Demonstration of Marangoni flow and sequential snap shots shows a micro particle transferring from NaOH to HCl. The tangential skin flow displaced component, as a result of the Marangoni effect, contains a Galinstan layer near the surface of the droplet, an oxide layer and a layer of electrolyte also near the surface. A thickness of $\delta$ is used to define the effective thickness of this layer. (**c**) Selected enlarged images showing droplet deformation (in black arrows) towards NaOH, while Marangoni flow (in red arrows) direction is towards HCl. (**d**) Reference diagram of liquid metal droplet dynamics under a pH imbalance. Each 'black square' or 'red circle' presents an experiment with measurable deformation ratio or Marangoni flow, respectively (overlapped 'black square' and 'red circle' indicates experiments with both measurable Marangoni flow and deformation ratios without a distinct dominating effect), in various ionic concentrations of HCl and NaOH between 0.3 and 3 mol l$^{-1}$ (pH $\sim 0.5$ to $\sim -0.5$ for HCl and $\sim 13.5$ to $\sim 14.5$ for NaOH, respectively). The background of the reference diagram is coloured blue, green and yellow accordingly, to represent each of the regions (deformation, deformation–Marangoni and Marangoni regions) discussed in the text. Scale bars, 1 mm.

no apparent change in performance). Days of operation did not dampen the deformation effect. However, the longevity in the Marangoni region seems to be significantly shorter (less than several hours of perfect repeatability).

The difference can be associated to the fact that the skin formed on the surface is very thin during the deformation process (several nanometres) and consequently, a very small amount of Galinstan is chemically used in the process. However, in the Marangoni flow region the surface is much thicker, delaminates and leaves the droplet, which, in turn, reduces the mass of the droplet.

To further examine the effect that the presence of a surface skin has on the establishment of the Marangoni flow, a set of experiments were conducted using mercury droplets (Supplementary Fig. 3). For mercury, hardly any surface oxide skin was observed after exposure to acidic and basic electrolytes. Interestingly, no Marangoni flow was observed for mercury droplets at the same concentrations where strong Marangoni flow was seen for Galinstan droplets.

Experiments were conducted to observe the effects of the changes in the differential pH across the droplet. Figure 3a,b illustrate the deformation and Marangoni flow of the droplet when the pH was kept constant in one of the channels (a concentration of $2.4 \, mol \, l^{-1}$ produces a solution with a pH $\sim -0.4$ and $\sim 14.4$ for HCl and NaOH, respectively) and the molarity of the electrolyte was changed in the other channel.

A concentration of $2.4 \, mol \, l^{-1}$ was chosen, as it covers a typical band for the three regions shown in Fig. 2c. The experiments indicate that the pH gradient has a significant effect on both deformation and Marangoni flow (Supplementary Movie 1). Within region 2, the deformation ratio decreases almost linearly when the molarity is increased (black line in Fig. 3a,b) and at the same time, the Marangoni flow rate increases (red line). Additional measurements on other acid and base types (for example, $H_2SO_4$ and KOH) are presented in Supplementary Fig. 4.

**Effect of mixed salt concentration on liquid metal droplet dynamics.** In these experiments, the change of maximum surface tension at the PZC is not separable from the effect of the pH change. To further understand the effect of the maximum surface tension at the PZC change and the accumulated charges in the EDL, several experiments were conducted by keeping both the acid and base concentration at set points, while changing the ionic concentration via the addition of a neutral salt. The salt chosen here was NaCl, which completely ionizes in water. The outcomes of these measurements are presented in Fig. 3c–f, Supplementary Fig. 5 and Supplementary Movie 2, in which the effect of NaCl concentration is shown at constant sets of acidic and basic concentration in the respective channels.

At $0.6 \, mol \, l^{-1}$ NaOH and three different concentrations of HCl (0.6, 1.2 and $2.4 \, mol \, l^{-1}$), deformation increases almost linearly by increasing [NaCl] from 0.3 to $1.2 \, mol \, l^{-1}$ (Fig. 3c,d). The deformation of the droplet reached a saturation value after this point. Interestingly, the highest extent of deformation was seen at $1.2 \, mol \, l^{-1}$ HCl. When the NaOH concentration was increased to $1.2 \, mol \, l^{-1}$, significant deformation of the droplet was already occurring at even the lowest concentration of NaCl and the changes were comparatively less prominent.

As mentioned in the previous section, for HCl concentrations of $< 0.6 \, mol \, l^{-1}$, the Marangoni flow is negligible. At higher concentrations of HCl, the Marangoni flow consistently shows a distinct peak when the NaCl concentration changes. Interestingly, the location of this peak is a function of the NaOH concentration; however, it is not affected by a change of NaCl concentration. After this peak, the Marangoni flow decreases. It seems that at high NaCl concentrations, the mixing may be more extensive and as a result, more NaCl is transferred to the HCl channel reducing the flow. There can also be a competition between the adsorption of $Cl^-$ and $OH^-$ ions. The $OH^-$ ions change the pH of the solution, which alters the voltage gradient that can adversely affect the adsorption of $Cl^-$ ions.

For HCl kept at three constant values of 0.6, 1.2 and $2.4 \, mol \, l^{-1}$ and when NaOH is at $2.4 \, mol \, l^{-1}$ (Fig. 3e,f),

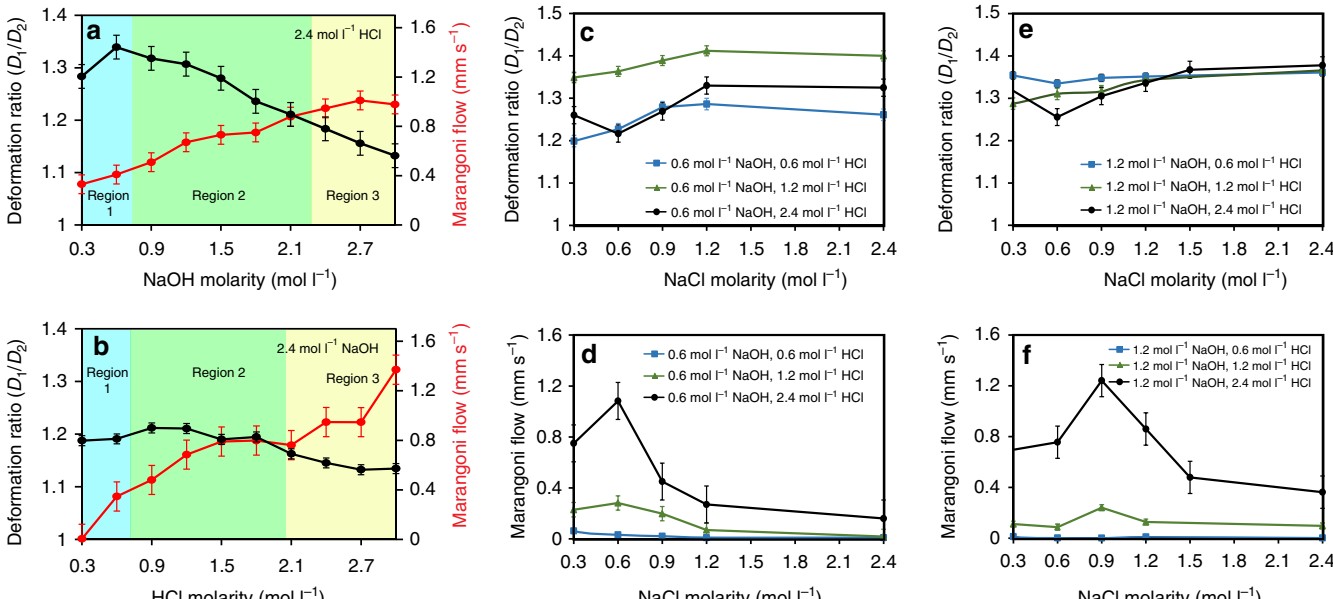

**Figure 3 | Changes of deformation and Marangoni flow under different conditions.** (**a**) Graph of the experimental measurements with varying NaOH and constant HCl molarities. Error bars are s.e.m. ($N = 6$). (**b**) Graph of the experimental measurements with varying HCl and constant NaOH molarities. Black and red lines indicate droplet deformation ratio and Marangoni flow rates, respectively. Background colours correspond to regions of the reference diagram in Fig. 2c. Error bars are s.e.m. ($N = 6$). (**c-f**) Graphs present the deformation ratio and Marangoni flow in varying concentrations of NaCl, while concentrations of NaOH and HCl are kept constant. NaCl is mixed with NaOH in all experiments. Error bars are s.e.m. ($N = 6$).

deformation starts at a higher magnitude in the non-saturated regions. The behaviour of Marangoni flow is similar to $0.6 \, mol \, l^{-1}$ NaOH but the peak is shifted towards higher NaCl concentrations.

If NaCl is added to HCl no deformation and Marangoni flow is observed. It seems that the competition between $Na^+$ and $H^+$ is counterproductive for both effects. Unfavourable effects of salt mixtures with some electrolytes on the EDL has been explained previously by Lyklema[50] and Davies[51].

**Voltage characterizations**. To verify and characterize the effect of the maximum surface tension at the PZC, the dynamics of droplets are investigated by applying an external electrical potential. Changing the potential changes the potential difference gradient across the EDL along the interface between the droplet and the electrolyte. This changes the surface tension gradient and as a result, both the deformation and Marangoni flows are altered. The set-up is shown in Fig. 4a. Both electrodes are Galinstan amalgamated metals that cannot be deformed. The EDL is manipulated by the applied voltage and its effect on the maximum surface tension can be assessed by the observation of deformation and surface flow (Fig. 4b and Supplementary Movie 3).

The channels were filled with acidic, basic and salt solutions of the same concentrations, to analyse the effect of applied voltage in different electrolytes. The measurement outcomes are presented in Fig. 4c,d. Five different conditions were shown for comparison. Molar concentrations of $0.6 \, mol \, l^{-1}$ NaOH and $1.2 \, mol \, l^{-1}$ HCl are shown to exhibit maximum deformation and relatively small Marangoni flow (Fig. 3a–d), which was confirmed by experiments in Fig. 4c,d.

When the applied voltage is increased, the deformation ratios reach peaks of > 1.25 in all cases. The peak reaches the maximum at a voltage of 1.5 V for HCl only, whereas the peak is significantly shifted to 2.1 V for NaOH only. Adding NaCl to HCl and NaOH shifts the peaks to the middle area between these two voltages to 1.7 and 1.9 V, respectively. This means the location of the

maximum surface tension at the PZC is influenced by mixing salt ions which interestingly, shifts both the HCl and NaOH peaks almost symmetrically. The order of the appearance of the peaks shown in Fig. 4c is in agreement with observations by Grahame on mercury[47]. The peak voltage shifts are also comparable with what is seen for mercury[47,52] and liquid gallium[48].

Marangoni flow velocity increases with an increase in the applied voltage and has a different set point for each electrolyte. The set point is lower for HCl (0.8 V) and significantly increases for NaOH (1.7 V). Similar to the deformation curves, solutions containing NaCl shift the Marangoni flow velocity to the middle, to 1 and 1.6 V for HCl and NaOH solutions, respectively that contain NaCl. Marangoni flow velocity is much higher for HCl, almost an order of magnitude larger than that of NaOH.

Assuming a constant differential EDL capacitance, the surface tension of Galinstan (Fig. 4e) can be readily obtained by applying the integrated Lippmann equation (equation (1)) at each PZC of the ionic solutions. The peak locations are obtained from experimental measurements. As can be seen, the calculated surface tension changes are proportionally related to the deformation ratio changes as a function of voltage, which is the expected observation. The parameters used are presented in Supplementary Table 1.

For the deformation region, the capacitance energy of the droplet due to ionic imbalance over the two sides of the droplet converts into mechanical energy, which induces droplet propulsion and tangential skin flow. The capacitance energy can be obtained using the integrated Lippmann equation (equation (1)) at each side of the droplet, as:

$$E_{\text{Unit area}} \cong \frac{1}{2} \left[ \underbrace{C_a (\varphi_a - \varphi_{oa})^2 A_a}_{\text{Acid}} + \underbrace{C_b (\varphi_b - \varphi_{0b})^2 A_b}_{\text{Base}} \right] \bigg/ A_{\text{Total}}$$

(3)

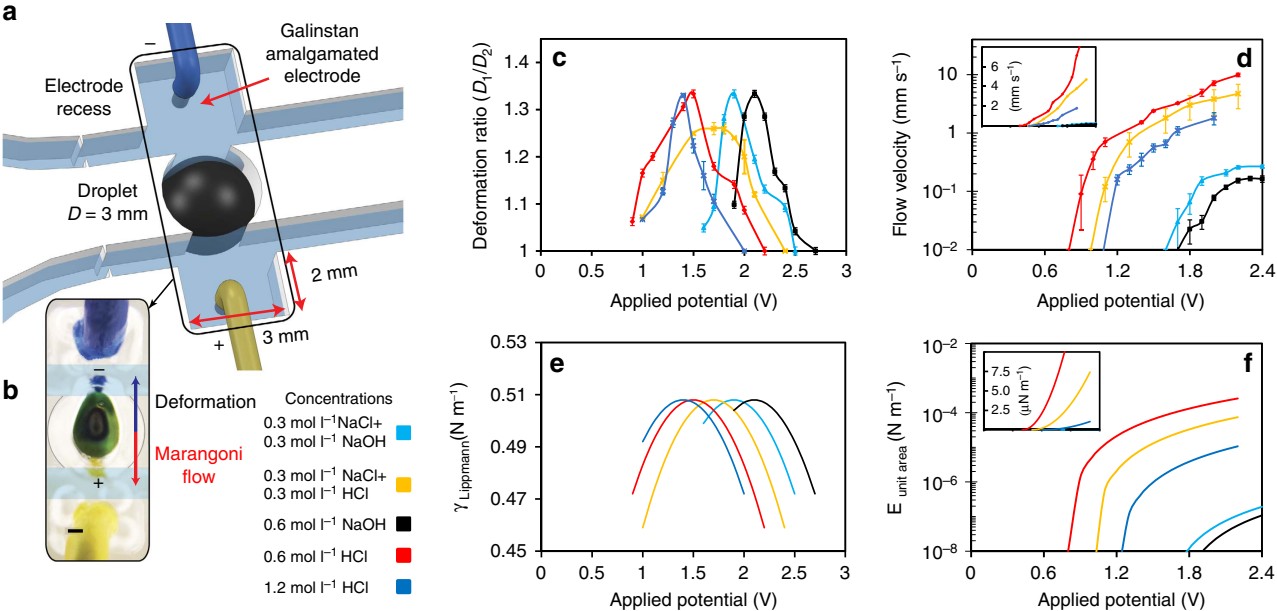

**Figure 4 | Characterization of droplet dynamics with an external applied electrical potential.** (**a**) Electrodes are covered with Galinstan and placed in rectangular recesses. (**b**) Marangoni flow and deformation direction of droplet created by the electrical potential. Scale bar, 0.5 cm. (**c**) Deformation ratios under the applied electrical potential. Error bars indicate s.e.m. (N = 6). (**d**) Marangoni flow velocities under the applied electrical potential. Applied potentials that are greater than presented here cause the oxidation of the liquid metal. Error bars indicate s.e.m. (N = 6). Calculated: (**e**) surface tension of the Galinstan droplet using Lippmann's equation (PZC value for EGaIn has been used in the simulation as the PZC of Galinstan[27]) and (**f**) the kinetic energies of Marangoni flows per unit area of a surface's cross-section. Insets in **d** and **f** show graphs in linear scales.

where $\varphi_i$ is the applied surface potential, $\varphi_{i0}$ is the PZC, $A_i$ is the surface area and $C_i$ is the capacity per surface area of one droplet semi-sphere. Here, $A_{Total}$ is the total surface area of the droplet and $i = a$ or $b$ denote values for acidic and basic electrolytes, respectively.

The equation describes that the two sides of the droplets can be independently affected by changing the acid and base concentrations. The asymmetry breaking is then a function of the total EDL energy of the two sides.

For the Marangoni region, Fig. 4f is the simulated representation for the kinetic energy of Marangoni flow per unit area. This is obtained using the following equations:

$$E_{Unit\ area} \cong \frac{1}{2} \int d\left(\frac{m_s U_s^2}{A_s}\right) \tag{4}$$

$$E_{Unit\ area} \cong \frac{1}{2}\delta\rho_s\alpha^2(\varphi - \varphi_{Th})^2 \text{ for } \varphi > \varphi_{Th} \tag{5}$$

Here $m_s$ is the displaced mass (this mass is made of surface liquid of the Galinstan, liquid in its vicinity and delaminated skin oxide layer in Fig. 2b and Supplementary Fig. 2c), $U_s$ is the skin tangential flow velocity generated by the Marangoni effect, $A_s$ is the surface area of the moving layer, $\rho_s$ is the density of the displaced mass, $\delta$ is the effective thickness of the displaced layer (Fig. 2b) at the interface of liquid metal (which also includes the surface oxide skin), $\alpha$ is the ionic liquid constant (which is a function of gradient of Marangoni flow against potential; Supplementary Table 1), $\varphi$ is the applied external potential and $\varphi_{Th}$ is the threshold for the applied external potential for the ionic liquid in which Marangoni flow starts. The discussion regarding the derivation of this equation is presented in the Supplementary Note 1. It is important to consider that this equation can also be extracted from the general Lippmann equation.

The main observation is that after fitting the voltage thresholds, Marangoni flow velocity graphs and the calculated kinetic energies trends are proportionally related.

**Applications of liquid metal based on ionic imbalance self-propulsion**. To demonstrate the self-propulsion of the liquid droplets due to pH and ionic concentration differences, selected regions based on Figs 2 and 3 were used to assess the effect of deformation and Marangoni flow on the behaviour of the droplets. A number of experiments were conducted and as a typical example, Fig. 5a presents images for the continuous motion of a self-propelling droplet in an open-top semi-cylindrical channel taken at different time intervals. The cylindrical shape of the channel allowed the spherical droplet to completely separate the two electrolytes. The tube was coated with Teflon to reduce the friction force. The droplet was placed midway between the two reservoirs and the channels were filled, as presented in Supplementary Movies 4 and 5. Figure 5b shows the profile of the droplet velocity in a channel length of 8.7 cm (Supplementary Movie 4). With the experimental conditions of 1.2 mol l$^{-1}$ HCl and 0.6 mol l$^{-1}$ NaOH, (which we later show to be the optimum conditions for self-propulsion), the droplet travelled at a maximum velocity of 25 mm s$^{-1}$ (Fig. 2b). Supplementary Fig. 6 and Supplementary Movie 4 show other experimental set points with velocities exceeding 20 mm s$^{-1}$. It is important to consider that after reaching the maximum velocity it almost drops linearly due to the height difference between the two reservoirs, which is generated after the displacement of the liquid, generating a pressure variance against the motion.

A set of experiments with various pH differences and without any addition of NaCl were also undertaken. In general, the average velocity of the droplet increases with increasing the acid concentration (Fig. 5c) until reaching 1.2 mol l$^{-1}$ HCl

(pH $\sim -0.1$) and then decreases. The peak velocity was generally associated with NaOH at a concentration of 0.6 mol l$^{-1}$ (pH $\sim 13.8$). The maximum velocity, as represented in Fig. 5b, reached 25 mm s$^{-1}$.

Another set of experiments were conducted to assess the effect of the ionic concentration of NaCl on the droplet velocities. The NaCl concentrations were chosen based on the experiments presented in Fig. 3 and minima were seen in all graphs. At 1.2 mol l$^{-1}$ NaOH, the minima occurs at 0.9 mol l$^{-1}$ NaCl and for 0.6 mol l$^{-1}$ NaOH it takes place at 0.6 mol l$^{-1}$ NaCl. These NaCl concentrations are very consistent with the locations of the Marangoni flow peak positions, as seen in Fig. 3. This reduction in velocity seems to be strongly associated with the effect of mixing of the electrolytes promoted by the Marangoni flow that deteriorates the EDL on each side and hence reduces the breakage of symmetry in the environment around the droplet. The maximum velocity belongs to the imbalanced condition of a mixed solution of 0.6 mol l$^{-1}$ NaOH + 0.9 mol l$^{-1}$ NaCl on one side and 1.2 mol l$^{-1}$ HCl on the other side. The droplet is capable of self-propelling longer distances, as shown in Supplementary Fig. 7 and Supplementary Movie 5 in a serpentine channel with a total length of 18.3 cm.

As an example of practical applicability, this pumping effect is tested (Fig. 5f). The liquid metal droplet is placed in a 1/8 inch diameter standard tubing and each side was filled with the acidic and basic electrolytes that gave the maximum velocity. As can be seen, the liquid droplet moved in < 1 s, generating a height difference of $\sim$9 mm, which accounts for $\sim$1 mbar pressure difference. This headspace pressure can be significantly increased by decreasing the surface area before entering the realm that atomic forces dominate the system.

Another example is the switching capability of the electrolyte only system (Fig. 5g), with the liquid metal droplet blocking access to the middle liquid reservoir. After electrolytes containing acid and base are introduced, the liquid droplet moves towards the basic area, opening the outlet of the middle reservoir. As a result, the middle liquid reservoir (containing dionised (DI) water/yellow food dye) is gradually mixed with the content of the acid reservoir.

Electrical actuation of mercury has been shown by Kim et al.[41]. Similarly, electrical actuations has been shown for liquid alloys of gallium by Tang et al.[18] and Gough et al.[28]. Accordingly in comparison, the ionic imbalance can perform better, in terms of velocity, than electrical actuation for voltages < 4 V when the deformation ratio is < 1.5. However, significantly larger velocities can be obtained (up to five times larger than the ionic imbalance condition) using applied electrical potentials of 10 V or higher.

**Discussion**

According to our observations, the stored energy on the surface of a droplet, due to the ionic imbalance, can be converted into: (a) droplet propulsion kinetic energy for low concentration electrolytes in the deformation region or (b) tangential skin flow for electrolytes of high concentrations in the Marangoni region. In reality, each component also involves an extra term representing the viscosity losses (propulsion viscosity loss for the deformation region and flow viscosity loss for the Marangoni region). By ignoring these viscosity loss components, the equation describing the system can be simplified into:

$$\text{Kinetic energy} \cong \begin{cases} \frac{1}{2}m_D U_D^2 & \text{Deformation region} \\ \frac{1}{2}\delta\rho_s A_s U_s^2 & \text{Marangoni region} \end{cases} \tag{6}$$

in which $m_D$ is the droplet mass, $U_D$ is the average propulsion velocity of the droplet due to the deformation effect, $U_s$ is the skin tangential flow velocity due to the Marangoni flow, $\rho_s$ is the

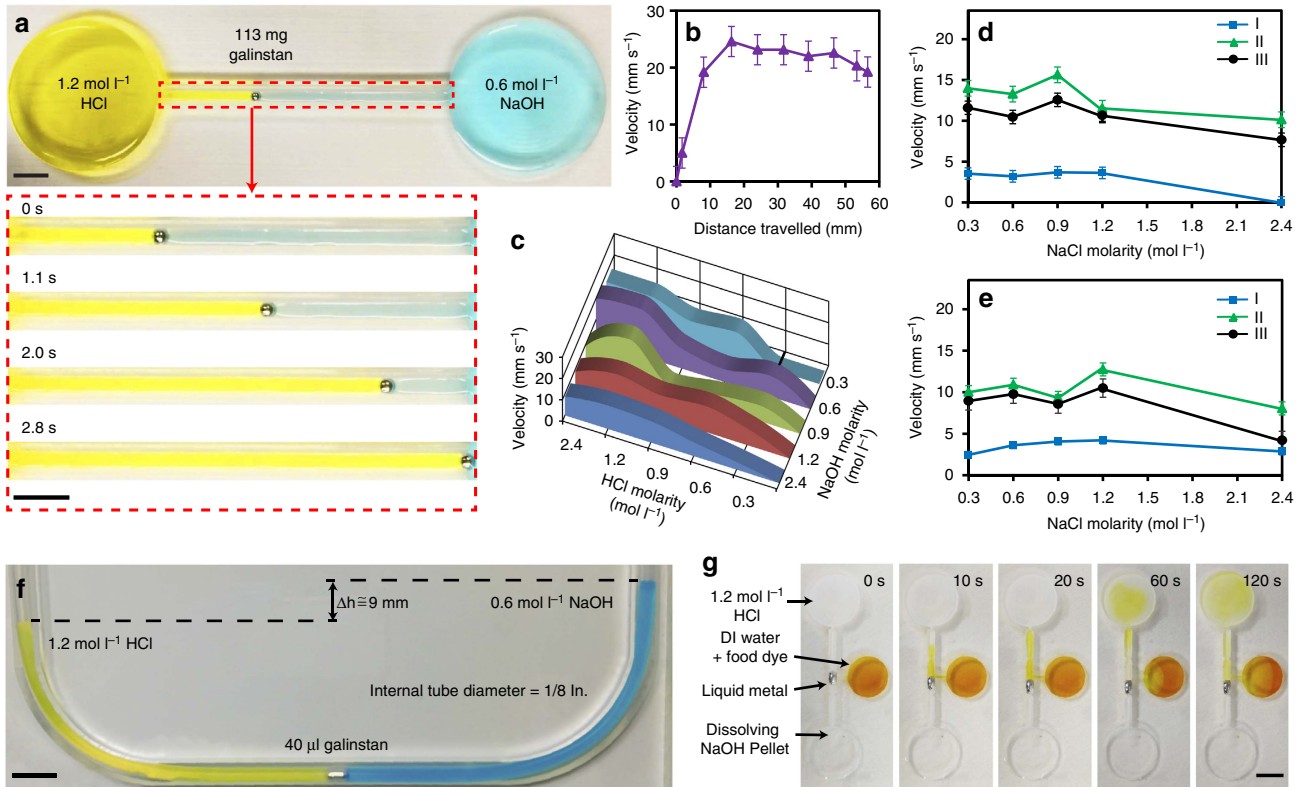

**Figure 5 | Liquid metal self-propulsion.** (**a**) Droplet propels from 1.2 mol l⁻¹ HCl to 0.6 mol l⁻¹ NaOH reservoir. (**b**) The instantaneous droplet velocity profile of experiment **a**. Error bars indicate the s.e.m. ($N = 2$). (**c**) Average velocity of droplets under different acidic and basic solutions. (**d,e**) Droplet velocities at different NaCl concentrations. Error bars indicate the s.e.m. ($N = 3$). I, II, III in **d** indicate constant 0.6 mol l⁻¹ NaOH (pH $\sim$ 13.8) and 0.6, 1.2 and 2.4 mol l⁻¹ HCl (pH 0.2, $\sim -0.1$ and $\sim -0.4$), respectively. I, II, III in **e** indicate constant 1.2 mol l⁻¹ NaOH (pH $\sim$ 14.1) and 0.6, 1.2 and 2.4 mol l⁻¹ HCl (pH $\sim$ 0.2, $\sim -0.1$ and $\sim -0.4$), respectively. (**f**) Metal droplet pushes the liquid to produce a 6.5 mm difference in height. According to the $\rho gh$ (in which $\rho$ is the density, $g$ is the gravitation acceleration constant and $h$ is the liquid height), this amount of liquid equates to a pressure exceeding $\sim$ 1 mbar for 1/8 inch diameter tubing. By reducing the area, this headpressure can be significantly increased. (**g**) The concept of a switch based on the motion induced by the electrolyte difference on either side of the droplet. The pH difference across the liquid metal droplet induces a motion towards the basic liquid and opens the inlet to the liquid with DI water/yellow dye after 10 s. The liquid in the middle reservoir then mixes with the acidified liquid in the top reservoir. Snapshots taken at: 0, 10, 20, 60 and 120 s. Scale bars, 1 cm.

density of the oxide skin layer moving tangentially along the surface of droplet, $\delta$ is the effective thickness of the skin layer (Fig. 2b) and $A_s$ is the average surface area of the skin layer. The kinetic energy of the liquid metal droplet, which is generated by the ionic imbalance, can be exploited according to equation (6).

pH and ionic concentration differences govern the components of equation (6) by determining the potentials and adsorbed surface charges. Exposure of the liquid metal droplet to a pH imbalance and ionic concentration differences induces a surface tension gradient across the droplet by (a) changing the PZC and (b) a generated potential. As demonstrated, in equation (6), the deformation component becomes dominant at relatively low pH values and low concentration differences. This is due to the fact that the self-limiting oxide surface in such conditions is atomically thin and the layer easily breaks down to allow the large deformation of the droplet (Fig. 2a and Supplementary Fig. 8). Conversely, at relatively high pH and concentration differences, the Marangoni flow becomes the dominant component. This is the region in which the oxide becomes thick enough to reduce the deformation effect (Fig. 2b and Supplementary Fig. 2).

The reader should consider equation (6) as valid under the conditions where there is no significant mixing of the electrolytes on the two sides of the droplet, due to the Marangoni flow. Obviously, in such an extreme regime, no symmetry breaking occurs and, as such, no droplet motion takes place.

An interesting observation regarding the maximum velocity of the droplets is about the surface tension of Galinstan. The maximum velocity of the Galinstan droplet is obtained ($\sim$ 25 mm s⁻¹) for 0.6 mol l⁻¹ NaOH (pH $\sim$ 13.8) and 1.2 mol l⁻¹ of HCl (pH $\sim -0.1$) condition (Supplementary Fig. 9). These two numbers are associated with the local maximum of surface tensions for the area in contact with acidic and the local minimum of the surface tension for the area in contact with basic electrolytes that probably generate the maximum surface tension difference. This consequently generates the highest asymmetry across the liquid metal droplet, resulting in the maximum velocity. However, when the oxide becomes thick enough on both sides of the droplet, the deformation effect is reduced. As shown in Supplementary Fig. 9, there still exists a significant surface tension gradient between the two sides of a droplet at high concentrations of acid and basic ionic electrolytes, despite the formation of this thick hydrated oxide layer (surface tension measurement calculations are presented in Supplementary Note 2 and Supplementary Table 2). Owing to the rigidity of the layers, this surface tension appears as Marangoni flow from NaOH towards HCl on the surface of the Galinstan. Parts of the skin layer are also delaminated and move with the tangential flow. This effect, however, is not seen for mercury droplets where an oxide layer is harder to form chemically (Supplementary Fig. 3).

In brief, in this study we have presented a comprehensive study of the autonomous motion of liquid metal droplets caused by

modification of the liquid electrolyte surrounding the liquid metal. The maximum velocity is obtained at the largest magnitude of induced surface tension gradient according to Lippmann's equation. The ionic properties of the electrolytes contain sufficient energy to induce motion of the metal droplet through modification of the EDL. Consequently, the liquid metal droplet is continuously propelled without the need for an external electric potential. We presented the measurements that showed some of the optimum conditions for gaining the maximum velocities of liquid metal droplets in an electrolyte only system. To present some of the applications of this only electrolyte-based liquid metal motion, we demonstrated the pumping capability of the system as well as its switching capability. Our work fundamentally presents the governing factors in obtaining the best condition for self-propulsion that can also be extended to voltage-induced motion systems. The outcomes of this work can be used for designing future autonomous low-dimensional micromechanical units that are based on compositional changes of the employed electrolytes, thereby exploiting the use of systems that only contain liquid components.

## Methods

**Ionic imbalance framework.** For the experiments presented in Figs 1–3, a Fusion 720 dual syringe pump was used to dispense acid and base at a steady flow rate of $200\,\mu l\,min^{-1}$ and a PHD 2,000 Harvard syringe pump was used to infuse at a steady flow rate of $400\,\mu l\,min^{-1}$. The effect of flow viscosity on droplets is assumed negligible. Inlet flow was regulated through a narrow orifice opening fabricated to avoid any inlet flow irregularities before sweeping across to the droplet. Video recording was analysed with a developed image processing programme in Matlab programme, to facilitate assessing the dynamics of the droplet. Cone-shaped ultraviolet and blue light wavelength band-pass filters attached to a camera lens to enclose droplet and mitigate light energy absorption. Photos and video recordings have a green reflection of the droplet surface facing the camera.

**Voltage characterization.** In Fig. 4, the recess and its walls are coated with Teflon to form a hydrophobic surface. Hydrophobicity avoids wetting of the intermediate surface by ionic solutions, which are an electrical conductor, that can bypass the EDL capacitor. Copper electrode tips are amalgamated with Galinstan to avoid generation of battery cells in between the droplet and electrodes. The two channels are completely segregated, to ensure the voltage drop occurs only on the liquid metal droplet. Electrolyte levels inside each channel were identical and lower than the droplet surface.

**Droplet self-propulsion.** In Fig. 5, a perfluoroalkoxy tube is cut in half using a milling machine to make a semi-cylindrical channel. Next, the perfluoroalkoxy semi-cut tube is coated with dry-film Teflon spray, to bring down the friction force and enhance the mobility of the droplet. Each reservoir is filled with 9 ml of solution. Galinstan liquid metal (113 mg) is separated into two smaller droplets and placed inside the channel. Next, each droplet is connected to either of the reservoirs. The droplets are then merged. Separation of the droplets minimizes mixing of the ionic solutions at the start of the experiments.

**Data availability.** The data that support the findings of this study are available from the corresponding authors upon request.

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

## Acknowledgements

A.C. acknowledge the support of the Victorian Government through the 2015 Victorian Postdoctoral Research Fellowship programme. The authors would also like to thank Micro Nano Research Facility (MNRF) at RMIT University.

## Author contributions

A.Z., T.D., K.K. and K.K.Z. conceived and designed the experiments. A.Z., T.D., A.C., J.Z.O. and K.K.Z. performed the experiments. All authors co-wrote the manuscript.

## Additional Information

**Competing financial interests:** The authors declare no competing financial interest.

