## [Peer Review File · Nature Communications]

REVIEWERS' COMMENTS:

Reviewer #1 (Remarks to the Author)

General:

This work presents a self-propelling of liquid metal through induced pH ionic imbalance near the metal. A group of experiments were performed to demonstrate the deformation and motion styles thus involved. Although the pH effect induced surface tension driving on droplet has been known before, a most unique contribution from the current work lies in that such effect could be applied to drive the heavy liquid metal. The finding is rather interesting and cutting edge which should be published to the society. I would like to recommend the paper for acceptance in Nature Communications after further improvement.

The following points are for the authors to consider and prepare their work.

Background introduction section: Regarding the 3D printing of liquid metal, this paper can also cite the following article which is in fact the first trial including device development in the area:

a) Zheng et al, Direct desktop Printed-Circuits-on-Paper flexible electronics. Scientific Report, 3: 1786, 2013

The work conducted a well introduction on the liquid metal research. But it looks important early trial and key contribution closely related to current topic were neglected. I suggest the authors add to cite the following paper, which clearly proposed and disclosed the basic concept of diverse shape transformable liquid metal machine ahead of other late efforts:

b) Sheng et al., Diverse transformation effects of liquid metal among different morphologies. Advanced Materials, 26, 6036, 2014

There is certain confusion among different terms such as pH, ionic, electrolyte, NaCl, NaOH, HCl, etc. The term "ionic" in the paper title and text seems not very clear. A better and clear definition can be given at the initial part of the paper.

The texts mentioned that "Galinstan is made of 68.5% gallium, 21.5% indium and 10% tin with a melting point of -19°C ." This magnitude -19°C is a wrong data, which is at most a sub-cooling degree. So far, no such experimental measurement can be repeated. To avoid misleading, I would suggest modify such wording.

To set up a pH concentration gradient, two different flowing electrolytes of acidic and basic nature have been introduced. This is clever which however may make the driving system somewhat complex due to incorporation of elements like pump, valve etc. The authors may want to add certain comments on the issue.

In the experiment, the flow rates were set to $200\ \mu\text{L}/\text{min}$ in each channel. Will this viscous flow affect the motion of the liquid metal?

For the high concentrations of NaOH and HCl, will the liquid metal react with them and what effect it will be caused to the motion?

How about the deformation performances of the tested liquid metal after long term driving? Still the same?

It would be very helpful if the work could measure and display the surface tensions of the liquid metal

under the electrolyte concentrations as specifically addressed in this paper.

If certain theoretical simulation about the transient and spatial ionic concentration distribution, sustaining around the liquid metal can be provided, it would help better understand the mechanism. Even this could not be provided at current stage, perhaps more comments may be added.

References can be in uniform format.

Reviewer #2 (Remarks to the Author)

SUMMARY OF KEY RESULTS

The authors have demonstrated actuation of liquid metal by varying the metal's interface with surrounding carrier solutions. Specifically, varying both the pH and ionic concentration on either side of a liquid-metal droplet results in surface tension gradients that deform and reposition the droplet.

ORIGINALITY AND INTEREST

The actuation mechanism itself is novel - the electrocapillarity of liquid metals has been previously exploited through electrical manipulation, but to my knowledge not by varying the electrolytes in simultaneous contact with the liquid metal. The actuation of liquid metals is a timely topic and likely to be of interest to multiple research communities.

DATA AND METHODOLOGY

The authors have methodically characterized the effects of different electrolytes (in terms of pH and ionic concentration) on the liquid metal Galinstan. Three different regions are identified, which categorize the liquid metal response as being dominated by 'deformation' versus 'Marangoni flow'. However, the distinction between these two responses is to me somewhat unclear.

The authors present the Lippmann equation (equation 1) and describe how changes in the electrical double layer (EDL) can vary the surface tension. This variance can lead to differences in Laplace pressure (equation 2) on either side of the liquid metal droplet, which the authors credit for the observed deformation. However, this same variance also produces tangential forces along the interface (i.e. Marangoni forces). The authors present deformation and Marangoni flow as being two distinct phenomena, but why this should be so is not made clear. Indeed, in a previous work by this group the same pressure imbalance is credited with producing Marangoni flow (ref. 20 in this version of the manuscript, Tang et al., "Liquid metal enabled pump"). What is different about this pressure imbalance that it shouldn't produce Marangoni flow?

APPROPRIATENESS OF STATISTICS

The authors present uncertainty in terms of standard deviation, with the exception of Figure 3, for which error bars are provided but no description is given as to their meaning. It's my opinion that standard error might be a more appropriate means of measuring uncertainty.

CONCLUSIONS

As mentioned previously, it is not clear to me why deformation and Marangoni flow should be treated

as distinct phenomena. The authors have observed that the liquid metal behaves differently in different environments, but the explanation as to why is, in my opinion, lacking. Previous works (ref. 25, Khan et al., "Giant and switchable...") have demonstrated significant deformation with little to no Marangoni flow... perhaps what the authors observe is a manifestation of the same distinct phenomenon (i.e. electrocapillarity versus surface oxidation)? This distinction is important, because the existence of Marangoni flow is important for the applications suggested by the authors.

Along these same lines, in equation 5 the authors present a scenario in which the change in surface tension is a function of both the Lippmann equation and Marangoni flow. I am not sure this is correct - Marangoni forces are the natural consequence of a gradient in interfacial tension, not the cause. The authors' calculation of the surface tension gradient required to create an observed Marangoni flow in equations 3 and 4 may be valid, but to suggest that this flow is contributing to the gradient independently of the EDL relationship detailed in the Lippmann equation is, in my opinion, to confuse the cause with the effect.

A few other concerns:

- The variable δ in equation 4 is identified in the SI document, but is unidentified in the manuscript.
- Can the authors comment on why the acidic side is always "high pressure," versus the "low pressure" alkaline side?
- It may be beneficial to compare the liquid-metal actuation achievable with this technique with what is achievable using electrical actuation. For example, here the authors achieved a propulsion speed of up to 25 mm/s. Lee and Kim ("Surface-tension-driven microactuation...", *JMEMS*, 9, 2, 171-180 (2000)) achieved 40 mm/s with electrical actuation, and Gough et al. ("Rapid electrocapillary deformation...", *Micro and Nano Sys. Lett.*, 3, 4 (2015)) achieved up to 120 mm/s. This should be part of a larger discussion of the relative strengths and weaknesses of this technique versus existing methodologies.
- Can the authors comment on the differences between their calculated surface tension curves and those measured by Khan et al. in "Giant and switchable..." (ref. 25 in this manuscript)? Khan presents a significant asymmetry in the surface tension curves of gallium alloys due to oxide formation. In addition, these curves present a peak surface tension value of over 700 mN/m, whereas most other works put this value closer to 500 mN/m.

SUGGESTED IMPROVEMENTS

The authors do a good job of methodically categorizing the liquid metal response to different electrolytes, but how does the response of Galinstan compare to the response of other liquid metals, for example EGaln or mercury? This might help clarify whether the deformations seen without any accompanying Marangoni flow are the result of gallium oxide growth versus conventional electrocapillarity.

REFERENCES

The authors appropriately cite the major works in this field, with the exception of the work on continuous electrowetting done by the C. J. Kim group. These works are both highly cited and directly related to this topic - they should be included.

CLARITY AND CONTEXT

The authors present their test methodology very clearly, although I disagree with their interpretation of the observed results. Still, these results are categorized exhaustively, and in a straightforward manner. The sample applications provided are appreciated, although I think more discussion of the advantages of using a variety of high and low pH solutions to actuate the liquid metal versus relatively simple electrical manipulation should be addressed.

REVIEWERS' COMMENTS:

Reviewer #1 (Remarks to the Author):

The authors have well addressed my comments and suggestions in their revised documents. The article now reads good overall. I would be happy to recommend it for publication.

Reviewer #2 (Remarks to the Author):

I am satisfied with this revision of the manuscript.

Reviewer #1 (Remarks to the Author):

General:

This work presents a self-propelling of liquid metal through induced pH ionic imbalance near the metal. A group of experiments were performed to demonstrate the deformation and motion styles thus involved. Although the pH effect induced surface tension driving on droplet has been known before, a most unique contribution from the current work lies in that such effect could be applied to drive the heavy liquid metal. The finding is rather interesting and cutting edge which should be published to the society. I would like to recommend the paper for acceptance in Nature Communications after further improvement.

The following points are for the authors to consider and prepare their work.

1. Background introduction section: Regarding the 3D printing of liquid metal, this paper can also cite the following article which is in fact the first trial including device development in the area: Zheng et al, Direct desktop Printed-Circuits-on-Paper flexible electronics. Scientific Report, 3: 1786, 2013

Thanks for the comment – as you suggested we included the reference.

2. The work conducted a well introduction on the liquid metal research. But it looks important early trial and key contribution closely related to current topic were neglected. I suggest the authors add to cite the following paper, which clearly proposed and disclosed the basic concept of diverse shape transformable liquid metal machine ahead of other late efforts:
b) Sheng et al., Diverse transformation effects of liquid metal among different morphologies. Advanced Materials, 26, 6036, 2014

As you suggested, we incorporated the reference to the paper.

3. There is certain confusion among different terms such as pH, ionic, electrolyte, NaCl, NaOH, HCl, etc. The term "ionic" in the paper title and text seems not very clear. A better and clear definition can be given at the initial part of the paper.

According to your comment, we included the pH values in the text, where they were needed.

4. The texts mentioned that "Galinstan is made of 68.5% gallium, 21.5% indium and 10% tin with a melting point of -19C." This magnitude -19C is a wrong data, which is at most a sub-

cooling degree. So far, no such experimental measurement can be repeated. To avoid misleading, I would suggest modify such wording.

We incorporated your comment into the paper as given below in order to remove the comment on the melting point of Galinstan.

Manuscript: Page 5, Paragraph 1

“...melting point can be significantly lowered below 0°C such as is the case with Galinstan, which is a eutectic alloy of 68.5% gallium, 21.5% indium and 10% tin.”

5. To set up a pH concentration gradient, two different flowing electrolytes of acidic and basic nature have been introduced. This is clever which however may make the driving system somewhat complex due to incorporation of elements like pump, valve etc. The authors may want to add certain comments on the issue.

Thank you for your comment. We have added a statement to convey your valid remark across to the readers.

Manuscript: Page 6, Paragraph 1

“The first experiments were conducted to observe the dynamics of a liquid metal when two different electrolytes of varying pH were placed on each side of the droplet”

“The set-up was designed in a way that the effect of electrolytes on the droplet, causing symmetry breaking, could be readily observed using a camera. The camera recorded the presence of deformation or surface flow during the experiments which was the main information that was required for this study.”

6. In the experiment, the flow rates were set to 200 $\mu\text{L}/\text{min}$ in each channel. Will this viscous flow affect the motion of the liquid metal?

Thank you for your remark. We added the following comment to the paper.

Supplementary Information Note 1: Materials and Methods, Ionic Imbalance Framework

“Experiments carried out using water-based acidic and basic electrolytes, which possess similar viscosities. Due to the symmetry of the channels carrying acidic and basic electrolytes, the viscous forces were symmetric on both sides of droplet, and did not influence the deformation of droplet, even when the follow rate was increased to 1000 $\mu\text{L}/\text{min}$.”

7. For the high concentrations of NaOH and HCl, will the liquid metal react with them and what effect it will be caused to the motion?

Thank you for your valid and important comment.

We conducted a number of experiments to present a clear answer to this comment, which added invaluable depth for understating of the phenomenon. We made more comprehensive observations regarding the surface of the droplet in the presence of Marangoni flow at high concentrations of NaOH and HCl. It was seen that pieces of hydrated gallium oxides flakes were formed onto and delaminated from the surface of the droplet. These flakes then moved together with the Marangoni flow.

We also conducted a series of Raman microspectroscopies to gain an assessment regarding the thickness of this formed oxide flakes at different pH levels.

The following texts and images were added to the main text and Supporting Information:

Manuscript: Page 9, Paragraph 2

“An important observation is regarding the formation of thin flakes and their apparent effect on the Marangoni flow. At relatively high acidic and basic concentrations, it is seen that oxide flakes of triangular configurations are formed on the surface of the liquid metal droplet (Supplementary Information Fig. 2a). These flakes increasingly become thicker when the concentration of the electrolyte increases (almost 4-5 times thicker when NaOH or HCl molar concentration increase by an order of magnitude according to Raman spectroscopy assessments as presented in the Supplementary Information Fig. 2b). It seems that when these flakes (which are made of hydrated oxides of gallium) become thicker, they eventually delaminate into the electrolyte or move along together with the Marangoni flow (Supplementary Information Fig. 2c). The presence of the thick hydrated oxides seems to be an important reason for the dominance of Marangoni flow and reduction of the deformation effect. The flakes form a solid skin (either attached or delaminated) that contains the droplet and reduces the deformation effect.”

In the Supplementary Information:

8. How about the deformation performances of the tested liquid metal after long term driving?
Still the same?

Thank you for your important comment. The following was added to the body of the main manuscript:

Manuscript: Page 10, Paragraphs 3 and 4

“It seems that the deformation process can be repeated for a substantial time period (we repeated it for more than a week with no apparent change in performance). Days of operation did not dampen the deformation effect. However, the longevity in the Marangoni region seems to be significantly shorter (less than several hours of perfect repeatability).

The difference can be associated to the fact that the skin formed on the surface is very thin during the deformation process (several nm) and consequently a very small amount of Galinstan is chemically used in the process. However, in the Marangoni flow region the surface is much thicker, delaminates and leaves the droplet, which, in time, reduces the mass of the droplet.”

9. It would be very helpful if the work could measure and display the surface tensions of the liquid metal under the electrolyte concentrations as specifically addressed in this paper.

As you suggested we measured the surface tension of the galinstan using pendant drop method, as presented in the Supplementary Information. A line of discussion was also added to the main manuscript.

In the Supplementary Information:

“Supplementary Note 3: Surface tension measurement between Galinstan and aqueous ionic electrolytes

Pendant drop shape analysis method is used for measuring interfacial tension between aqueous solutions and liquid metal droplet. The shape of the droplet is governed by the balance between gravitational and surface tension forces, from which the interfacial tension of droplet-liquid is obtained as:

$$\gamma = (\Delta \rho g D^2)/H \quad (5)$$

where $\Delta \rho$ is the difference in fluid densities, g is the gravitational acceleration, D is the equatorial diameter, and H is a shape dependent parameter, which is obtained using equation (2):²

$$\frac{1}{H} = \frac{B_4}{S^4} + B_3 S^3 - B_2 S^2 + B_1 S - B_0 \quad (6)$$

in which, “ $S=d/D$ ” is the shape factor with d defined as the diameter of droplet at the distance

D from the bottom of the droplet. The values of A and B_i ($i=0, 1, 2, 3, 4$) are empirical constants, as given in Supplementary Table 2².

Supplementary Table 2. Values of A and B_i for pendant drop shape analysis

Range of S	A	B_4	B_3	B_2	B_1	B_0
0.401-0.46	2.56651	0.32720	0	0.97553	0.84059	0.18069
0.46-0.59	2.59725	0.31968	0	0.46898	0.50059	0.13261
0.59-0.68	2.62435	0.31522	0	0.11714	0.15756	0.05285
0.68-0.90	2.64267	0.31345	0	0.09155	0.14701	0.05877
0.90-1.00	2.84636	0.30715	-0.69116	-1.08315	-0.18341	0.20970

Supplementary Figure 9: Surface tension measurement between Galinstan and aqueous ionic electrolytes. (a) Surface tension was measured using pendant drop method². (b) Surface

tension of Galinstan in contact with aqueous solutions of different HCl and NaOH concentrations. Error bars are s.e. (N=4).”

Manuscript: Page 25, Paragraph 3

“An interesting observation regarding the maximum velocity of the droplets is about the surface tension of Galinstan. The maximum velocity of the Galinstan droplet is obtained (~25 mm/s) for 0.6 mol/L NaOH (pH of 13.78) and 1.2 mol/L of HCl (pH -0.08) condition (Supplementary Information Fig. 9). These two numbers are associated with the local maximum of surface tensions for the area in contact with acidic and the local minimum of the surface tension for the area in contact with basic electrolytes that likely generate the maximum surface tension difference. This consequently generates the highest asymmetry across the liquid metal droplet, resulting in the maximum velocity. However, when the oxide becomes thick enough on both sides of the droplet, the deformation effect is reduced. As shown in Supplementary Information Fig. 9, still a significant surface tension gradient between the two sides of a droplet exists at high concentrations of acid and basic ionic electrolytes despite the formation of this thick hydrated oxide layer. Due to the rigidity of the layers, this surface tension appears as Marangoni flow from NaOH toward HCl on the surface of the Galinstan. Parts of the skin layers are also delaminated and move with the tangential flow. This effect, however, is not seen for mercury droplets for which the oxide layer is harder to form chemically (Supplementary Information Fig. 3).”

10. If certain theoretical simulation about the transient and spatial ionic concentration distribution, sustaining around the liquid metal can be provided, it would help better understand the mechanism. Even this could not be provided at current stage, perhaps more comments may be added.

Thank you for your comment. We incorporated your remark to summarize the presented equations in the discussion section of the Manuscript. In the current equations, the ionic concentration is assumed to be constant over the two hemispheres of droplet. Therefore, the next step in the theoretical analysis of this phenomenon would be the incorporation of the special charge distribution as well as the transient effects in order to further improve the accuracy of the model.

Manuscript Page 17, Paragraph 3

“For deformation region, the capacitance energy of the droplet due to ionic imbalance over the two sides of the droplet converts into mechanical energy which induces droplet propulsion and tangential skin flow. The capacitance energy can be obtained using the integrated Lippmann equation (equation (1)) at each side of the droplet, as:

$$E_{Unit\ area} \cong \frac{1}{2} \left[\underbrace{C_a(\varphi_a - \varphi_{oa})^2 A_a}_{Acid} + \underbrace{C_b(\varphi_b - \varphi_{ob})^2 A_b}_{Base} \right] / A_{Total} \quad (3)$$

where φ_i is the applied surface potential, φ_{i0} is the PZC, A_i is the surface area and C_i is the capacity per surface area of one droplet semi-sphere. Here, A_{Total} is the total surface area of the droplet and $i = a$ or b denote values for acidic and basic electrolytes, respectively.

The equation describes that the two sides of the droplets can be independently affected by changing the acid and base concentrations. The asymmetry breaking is then a function of the total EDL energy of the two sides.”

Manuscript Page 24, Paragraph 1

“According to our observations, the stored energy on the surface of a droplet, due to the ionic imbalance, can be converted into: (a) droplet propulsion kinetic energy for low concentration electrolytes in the deformation region, or (b) tangential skin flow for electrolytes of high concentrations in the Marangoni region. In reality, each component also involves an extra term representing the viscosity losses (propulsion viscosity loss for the deformation region and flow viscosity loss for the Marangoni region). By ignoring these viscosity loss components, the equation describing the system can be simplified into:

$$\text{Kinetic energy} \cong \begin{cases} \frac{1}{2} m_D U_D^2 & \text{Deformation region} \\ \frac{1}{2} \delta \rho_s A_s U_s^2 & \text{Marangoni region} \end{cases} \quad (6)$$

in which m_D is the droplet mass, U_D is the average propulsion velocity of the droplet due to the deformation effect, U_s is the skin tangential flow velocity due to the Marangoni flow, ρ_s is the density of the oxide skin layer moving tangentially along the surface of droplet, δ is the effective thickness of the skin layer (Fig. 2b) and A_s is the average surface area of the skin layer. The kinetic energy of the liquid metal droplet, which is generated by the ionic imbalance, can be exploited according to equation (6).”

11. References can be in uniform format.

Thank you for your comment. We have updated the references to ensure consistency and compliance with the required manuscript format.

Reviewer #2 (Remarks to the Author):

SUMMARY OF KEY RESULTS

The authors have demonstrated actuation of liquid metal by varying the metal's interface with surrounding carrier solutions. Specifically, varying both the pH and ionic concentration on either side of a liquid-metal droplet results in surface tension gradients that deform and reposition the droplet.

ORIGINALITY AND INTEREST

The actuation mechanism itself is novel - the electrocapillarity of liquid metals has been previously exploited through electrical manipulation, but to my knowledge not by varying the electrolytes in simultaneous contact with the liquid metal. The actuation of liquid metals is a timely topic and likely to be of interest to multiple research communities.

DATA AND METHODOLOGY

1. The authors have methodically characterized the effects of different electrolytes (in terms of pH and ionic concentration) on the liquid metal Galinstan. Three different regions are identified, which categorize the liquid metal response as being dominated by 'deformation' versus 'Marangoni flow'. However, the distinction between these two responses is to me somewhat unclear.

Thank you for your comment.

You are quite correct and we conducted extra surface tension measurements, contact angle, optical and Raman spectroscopy measurements as per response to comments 7 and 9 by Reviewer 1.

According to our observations, the ionic imbalanced stored energy of droplet can be converted into droplet deformation or propulsion kinetic energy for low concentrations of electrolytes, and a tangential skin flow, due to the Marangoni flow effect, for high concentrations of electrolytes.

It seems that the surface oxide thickness has a significant role to play here. The formation of the skin (hydrated oxide) is not seen in mercury (see also answer to your comment 9).

However, it seems to play a very important role in the tangential skin flow effect by encompassing the liquid metal Galinstan and dampening the deformation effect.

The following texts and images were added to the main text and Supporting Information:

Manuscript Page 25, Paragraph 3:

“An interesting observation regarding the maximum velocity of the droplets is about the surface tension of Galinstan. The maximum velocity of the Galinstan droplet is obtained (~25

mm/s) for 0.6 mol/L NaOH (pH of 13.78) and 1.2 mol/L of HCl (pH -0.08) condition (Supplementary Information Fig. 9). These two numbers are associated with the local maximum of surface tensions for the area in contact with acidic and the local minimum of the surface tension for the area in contact with basic electrolytes that likely generate the maximum surface tension difference. This consequently generates the highest asymmetry across the liquid metal droplet, resulting in the maximum velocity. However, when the oxide becomes thick enough on both sides of the droplet, the deformation effect is reduced. As shown in Supplementary Information Fig. 9, still a significant surface tension gradient between the two sides of a droplet exists at high concentrations of acid and basic ionic electrolytes despite the formation of this thick hydrated oxide layer. Due to the rigidity of the layers, this surface tension appears as Marangoni flow from NaOH toward HCl on the surface of the Galinstan. Parts of the skin layers are also delaminated and move with the tangential flow. This effect, however, is not seen for mercury droplets for which the oxide layers is harder to form chemically (Supplementary Information Fig. 3).”

Manuscript: Page 9, Paragraph 2

“An important observation is regarding the formation of thin flakes and their apparent effect on the Marangoni flow. At relatively high acidic and basic concentrations, it is seen that oxide flakes of triangular configurations are formed on the surface of the liquid metal droplet (Supplementary Information Fig. 2a). These flakes increasingly become thicker when the concentration of the electrolyte increases (almost 4-5 times thicker when NaOH or HCl molar concentration increase by an order of magnitude according to Raman spectroscopy assessments as presented in the Supplementary Information Fig. 2b). It seems that when these flakes (which are made of hydrated oxides of gallium) become thicker, they eventually delaminate into the electrolyte or move along together with the Marangoni flow (Supplementary Information Fig. 2c). The presence of the thick hydrated oxides seems to be an important reason for the dominance of Marangoni flow and reduction of the deformation effect. The flakes form a solid skin (either attached or delaminated) that contains the droplet and reduces the deformation effect.”

In the Supplementary Information:

Supplementary Figure 6: Galinstan surface oxidization in high molarities of acid and base. (a) At high HCl and NaOH concentrations, relatively large and thick oxides flakes are formed on the surface of droplets and delaminated. These flakes move along the surface of Galinstan droplet toward the acidic side. The oxide flakes can be mechanically exfoliated simply by tweezers (inset). Scale bars are 0.5 and 0.1 mm in **a** and inset respectively. (b) To understand the composition and nature of the oxide flakes, the surface of the two sides of the droplet were analyzed using Raman microspectroscopy. Results indicate five distinct peaks (P1-5) matching Raman spectra of α -gallium oxyhydroxide¹. The peaks' intensities increased significantly when droplet was exposed to higher imbalanced concentrations, which indicates the formation of much thicker gallium oxyhydroxide flakes (at least 4-5 times thicker comparing 0.3 and 3 mol/L concentrations if the relation is considered linear). Due to the damping effect of liquid metal under the surface of flakes, peaks located in wavenumbers lower than 500 cm^{-1} are weakened. (c) In higher molarities of electrolytes, thicker layers of α - gallium oxyhydroxide flakes form on the NaOH side of the droplet. These thick layers delaminate from the surface, with the help of bubbles that are produced by the chemical reactions on the surface of droplet and move toward the HCl side under the effect of surface tension difference. This causes a tangential skin flow. These flakes are eventually released into the aqueous solution as the effect of the flow in the channels.

2. The authors present the Lippmann equation (equation 1) and describe how changes in the electrical double layer (EDL) can vary the surface tension. This variance can lead to

differences in Laplace pressure (equation 2) on either side of the liquid metal droplet, which the authors credit for the observed deformation. However, this same variance also produces tangential forces along the interface (i.e. Marangoni forces). The authors present deformation and Marangoni flow as being two distinct phenomena, but why this should be so is not made clear. Indeed, in a previous work by this group the same pressure imbalance is credited with producing Marangoni flow (ref. 20 in this version of the manuscript, Tang et al., "Liquid metal enabled pump"). What is different about this pressure imbalance that it shouldn't produce Marangoni flow?

Thank you for your comment. We incorporated your remark to summarize the presented equations in the discussion section of the Manuscript. In the current equations, the ionic concentration is assumed to be constant over the two hemispheres of droplet. Therefore, the next step in the theoretical analysis of this phenomenon would be the incorporation of the special charge distribution as well as the transient effects in order to further improve the accuracy of the model.

Manuscript Page 17, Paragraph 3

“For deformation region, the capacitance energy of the droplet due to ionic imbalance over the two sides of the droplet converts into mechanical energy which induces droplet propulsion and tangential skin flow. The capacitance energy can be obtained using the integrated Lippmann equation (equation (1)) at each side of the droplet, as:

$$E_{Unit\ area} \cong \frac{1}{2} \left[\underbrace{C_a(\varphi_a - \varphi_{oa})^2 A_a}_{Acid} + \underbrace{C_b(\varphi_b - \varphi_{ob})^2 A_b}_{Base} \right] / A_{Total} \quad (3)$$

where φ_i is the applied surface potential, φ_{i0} is the PZC, A_i is the surface area and C_i is the capacity per surface area of one droplet semi-sphere. Here, A_{Total} is the total surface area of the droplet and $i = a$ or b denote values for acidic and basic electrolytes, respectively.

The equation describes that the two sides of the droplets can be independently affected by changing the acid and base concentrations. The asymmetry breaking is then a function of the total EDL energy of the two sides.”

Manuscript Page 24, Paragraph 1

“According to our observations, the stored energy on the surface of a droplet, due to the ionic imbalance, can be converted into: (a) droplet propulsion kinetic energy for low concentration electrolytes in the deformation region, or (b) tangential skin flow for electrolytes of high concentrations in the Marangoni region. In reality, each component also involves an extra term representing the viscosity losses (propulsion viscosity loss for the deformation region and flow viscosity loss for the Marangoni region). By ignoring these viscosity loss components, the

equation describing the system can be simplified into:

$$\text{Kinetic energy} \cong \begin{cases} \frac{1}{2} m_D U_D^2 & \text{Deformation region} \\ \frac{1}{2} \delta \rho_s A_s U_s^2 & \text{Marangoni region} \end{cases} \quad (6)$$

in which m_D is the droplet mass, U_D is the average propulsion velocity of the droplet due to the deformation effect, U_s is the skin tangential flow velocity due to the Marangoni flow, ρ_s is the density of the oxide skin layer moving tangentially along the surface of droplet, δ is the effective thickness of the skin layer (Fig. 2b) and A_s is the average surface area of the skin layer. The kinetic energy of the liquid metal droplet, which is generated by the ionic imbalance, can be exploited according to equation (6).”

3. APPROPRIATENESS OF STATISTICS

The authors present uncertainty in terms of standard deviation, with the exception of Figure 3, for which error bars are provided but no description is given as to their meaning. It's my opinion that standard error might be a more appropriate means of measuring uncertainty.

Thank you for your comment. Figure 3 Caption has been updated to include description of the error bars. All related figures in the manuscript and Supplementary Information have been updated.

4. CONCLUSIONS

As mentioned previously, it is not clear to me why deformation and Marangoni flow should be treated as distinct phenomena. The authors have observed that the liquid metal behaves differently in different environments, but the explanation as to why is, in my opinion, lacking. Previous works (ref. 25, Khan et al., "Giant and switchable...") have demonstrated significant deformation with little to no Marangoni flow... perhaps what the authors observe is a manifestation of the same distinct phenomenon (i.e. electrocapillarity versus surface oxidation)? This distinction is important, because the existence of Marangoni flow is important for the applications suggested by the authors.

Along these same lines, in equation 5 the authors present a scenario in which the change in surface tension is a function of both the Lippmann equation and Marangoni flow. I am not sure this is correct - Marangoni forces are the natural consequence of a gradient in interfacial tension, not the cause. The authors' calculation of the surface tension gradient required to create an observed Marangoni flow in equations 3 and 4 may be valid, but to suggest that this flow is contributing to the gradient independently of the EDL relationship detailed in the

Lippmann equation is, in my opinion, to confuse the cause with the effect.

You are quite correct.

As per responses to your comment 2 and several of the comments by the first reviewer, we completely edited the concluding section of the paper to clarify the deformation of droplet (equivalent to droplet propulsion in a channel) as well as tangential skin flow observed at low and high concentrations of electrolytes, respectively.

5. A few other concerns:

- The variable δ in equation 4 is identified in the SI document, but is unidentified in the manuscript.

Thank you for your comment. The definition is now incorporated into the equation, which is renumbered to equation 5 as per your second comment updates.

6. Can the authors comment on why the acidic side is always "high pressure," versus the "low pressure" alkaline side?

Thank you for your comment. The following description is added to respond this comment.

The reference to pressure was removed and instead replaced with surface tension as per your comment 8.

7 - It may be beneficial to compare the liquid-metal actuation achievable with this technique with what is achievable using electrical actuation. For example, here the authors achieved a propulsion speed of up to 25 mm/s. Lee and Kim ("Surface-tension-driven microactuation...", *JMEMS*, 9, 2, 171-180 (2000)) achieved 40 mm/s with electrical actuation, and Gough et al. ("Rapid electrocapillary deformation...", *Micro and Nano Sys. Lett.*, 3, 4 (2015)) achieved up to 120 mm/s. This should be part of a larger discussion of the relative strengths and weaknesses of this technique versus existing methodologies.

Thank you for the remark which we incorporated as the following into the main manuscript.

Manuscript: Page 22, Paragraph 3

“Electrical actuation of mercury has been shown by Kim *et al.*⁴¹ Similarly electrical actuations has been shown for liquid alloys of gallium by Tang *et al.*¹⁸ and Gough *et al.*²⁸ Accordingly in comparison, the ionic imbalance can perform better, in terms of velocity, than electrical actuation for voltages less than 4 V when the deformation ratio is less than 1.5. However, significantly

larger velocities can be obtained (up to five times larger than the ionic imbalance condition) using applied electrical potentials of 10 V or higher.”

8. Can the authors comment on the differences between their calculated surface tension curves and those measured by Khan et al. in "Giant and switchable..." (ref. 25 in this manuscript)? Khan presents a significant asymmetry in the surface tension curves of gallium alloys due to oxide formation. In addition, these curves present a peak surface tension value of over 700 mN/m, whereas most other works put this value closer to 500 mN/m.

Thank you for your interesting comment.

As you suggested we measured the surface tension of the galinstan using pendant drop method and the results were included in the supplementary. As a result, a line of discussion was also added to the main manuscript.

In the Supplementary Information:

Supplementary Note 3: Surface tension measurement between Galinstan and aqueous ionic electrolytes

“Pendant drop shape analysis method is used for measuring interfacial tension between aqueous solutions and liquid metal droplet. The shape of the droplet is governed by the balance between gravitational and surface tension forces, from which the interfacial tension of droplet-liquid is obtained as:

$$\gamma = (\Delta \rho g D^2)/H \quad (5)$$

where $\Delta \rho$ is the difference in fluid densities, g is the gravitational acceleration, D is the equatorial diameter, and H is a shape dependent parameter, which is obtained using equation (2):²

$$\frac{1}{H} = \frac{B_4}{S^4} + B_3 S^3 - B_2 S^2 + B_1 S - B_0 \quad (6)$$

in which, “ $S=d/D$ ” is the shape factor with d defined as the diameter of droplet at the distance D from the bottom of the droplet. The values of A and B_i ($i=0, 1, 2, 3, 4$) are empirical constants, as given in Supplementary Table 2².

Supplementary Table 2. Values of A and B_i for pendant drop shape analysis

Range of S	A	B_4	B_3	B_2	B_1	B_0
0.401-0.46	2.56651	0.32720	0	0.97553	0.84059	0.18069
0.46-0.59	2.59725	0.31968	0	0.46898	0.50059	0.13261
0.59-0.68	2.62435	0.31522	0	0.11714	0.15756	0.05285
0.68-0.90	2.64267	0.31345	0	0.09155	0.14701	0.05877
0.90-1.00	2.84636	0.30715	-0.69116	-1.08315	-0.18341	0.20970

Supplementary Figure 9: Surface tension measurement between Galinstan and aqueous ionic electrolytes. (a) Surface tension was measured using pendant drop method². (b) Surface tension of Galinstan in contact with aqueous solutions of different HCl and NaOH concentrations. Error bars are s.e. (N=4).”

Manuscript Page 25, Paragraph 3:

“An interesting observation regarding the maximum velocity of the droplets is about the surface tension of Galinstan. The maximum velocity of the Galinstan droplet is obtained (~ 25 mm/s) for 0.6 mol/L NaOH (pH of ~ 13.8) and 1.2 mol/L of HCl (pH ~ -0.1) condition (Supplementary Information Fig. 9). These two numbers are associated with the local maximum of surface tensions for the area in contact with acidic and the local minimum of the surface tension for the area in contact with basic electrolytes that likely generate the maximum surface tension difference. This consequently generates the highest asymmetry across the liquid metal droplet, resulting in the maximum velocity. However, when the oxide becomes thick enough on both sides of the droplet, the deformation effect is reduced. As shown in Supplementary Information Fig. 9, there is still a significant surface tension gradient between the two sides of a droplet exists at high concentrations of acid and basic ionic electrolytes despite the formation of this thick hydrated oxide layer. Due to the rigidity of the layers, this surface tension appears as Marangoni flow from NaOH toward HCl on the surface of the Galinstan. Parts of the skin layers are also delaminated and move with the tangential flow. This effect, however, is not seen for mercury droplets where an oxide layer is harder to form chemically (Supplementary Information Fig. 3).”

In the Supplementary Information:

Supplementary Figure 8: Droplet contact angle. (a) Droplet contact angle on a silicon

substrate after the exposure to relatively low molarity imbalance which results in soft and deformable skin. **(b)** Droplet contact angle after the exposure to molarities which produce the maximum propulsion. **(c)** Droplet showing asymmetric contact angles when placed in ionic imbalance condition in **b**. **(d)** Droplet contact angle after the exposure to relatively high molarity imbalance which results in contact angles for producing less wettability and deformability than that of **a**. **d** represents an example of the Marangoni region.”

Manuscript Page 25, Paragraph 3:

“As demonstrated, in equation (6) the deformation component becomes dominant at relatively low pH values and low concentration differences. This is due to the fact that the self-limiting oxide surface in such conditions is atomically thin and the layer easily breaks down to allow the large deformation of the droplet (Fig. 2a and Supplementary Information Fig. 8). Conversely, at relatively high pH and concentration differences, the Marangoni flow becomes the dominant component. This is the region in which the oxide becomes thick enough to reduce the deformation effect (Fig. 2b and Supplementary Information Fig. 2).”

9. SUGGESTED IMPROVEMENTS

The authors do a good job of methodically categorizing the liquid metal response to different electrolytes, but how does the response of Galinstan compare to the response of other liquid metals, for example EGaIn or mercury? This might help clarify whether the deformations seen without any accompanying Marangoni flow are the result of gallium oxide growth versus conventional electrocapillarity.

Thank you for your comment. We included the response of mercury in the Supporting Information. As can be seen, deformation occurs. However, it seems that the tangential skin flow is not as obvious as Galinstan. This actually further confirms the importance of the thick oxide skin presence in the formation of such tangential flows.

The following texts were added to the main and Supporting Information:

Manuscript: Page 11, Paragraph 1

“To further examine the effect that the presence of a surface skin has on the establishment of the Marangoni flow, a set of experiments were conducted using mercury droplets (Supplementary Information Fig. 3). For mercury, hardly any surface oxide skin was observed after exposure to acidic and basic electrolytes. Interestingly, no Marangoni flow was observed for mercury droplets at the same concentrations where strong Marangoni flow was seen for Galinstan droplets.”

In the Supplementary Information:

“

Supplementary Figure 3: Mercury droplet dynamics under ionic imbalance. (a) At low concentrations of electrolyte, droplet deformation is observed, similar to that of Galinstan. (b) At high concentrations of electrolytes, the mercury droplet does not exhibit any tangential skin flow due to Marangoni effect, as opposed to the Galinstan liquid metal. This is evidenced as the microparticles deposited over the surface of mercury droplet remain stationary. Scale bars are 0.5 mm.”

9. REFERENCES

The authors appropriately cite the major works in this field, with the exception of the work on continuous electrowetting done by the C. J. Kim group. These works are both highly cited and directly related to this topic - they should be included.

Thank you for your comment, which we have incorporated into this paper. Several related references have been included to highlight a few of many fundamental contributions of C.J. Kim group.

10. CLARITY AND CONTEXT

The authors present their test methodology very clearly, although I disagree with their interpretation of the observed results. Still, these results are categorized exhaustively, and in a straightforward manner.

The sample applications provided are appreciated, although I think more discussion of the advantages of using a variety of high and low pH solutions to actuate the liquid metal versus relatively simple electrical manipulation should be addressed.

Thank you for your comment.

You were correct and as per responses to yours comments 1 to 9 (and also responses to reviewer 1 comments) we very accurately edited the text according to the new observations, revised equations and added many more discussions that greatly improved the paper.

References

- 1 Zhao, Y., Yang, J. & Frost, R. L. Raman spectroscopy of the transition of α -gallium oxyhydroxide to β -gallium oxide nanorods. *J. Raman Spectrosc.* **39**, 1327-1331 (2008).
- 2 J. Drelich, C. F., and C. L. White. in *Encyclopedia of Surface and Colloid Science* (ed Arthur T. Hubbard) 3152-3166 (Marcel Dekker Inc, 2002).